# Sustainable Communication; Perceived Motivation and Nature of the Commitment

Banu Dincer  and Caner Dincer *

Department of Business Administration, Faculty of Economic and Administrative Sciences,
Galatasaray University, Çırağan Cad. No: 36, Ortaköy, 34349 Istanbul, Turkey
* Correspondence: cdincer@gsu.edu.tr

**Abstract:** The aim of this study is to examine the overlap between sustainable communication and business, as well as its impact on the consumer's perception and the nature of the motivation. We attempted to create a synthesis of prior research based on a literature review to understand how consumers comprehend corporate sustainability communications. The selection of the articles and related works is based on the presence of the keywords in the Science Direct database. The results provided us with 15 results for the research papers with "sustainable communication" in the title and 46 results with the keyword "sustainable communication" in the title, abstract or author-specified keywords since the year 2000. We synthesized these works and major works in the field according to our model, based on the attribution theory frame. We have emphasized the effect of sustainable communication fit with the company on the consumer's perception of internal motivation by mobilizing the attribution and congruence theories, while highlighting the importance of the company's perceived effort and the moderating role of other situational variables.

**Keywords:** sustainable communication; perceived motivation; attribution theory; congruence theory



## 1. Introduction

In a context characterized by increased societal awareness, consumers are increasingly developing new requirements in this field. In return, companies are obliged to become heavily involved not only in social practices but also in the communication of their actions to stakeholders and the public. This has given rise to a new form of communication called "sustainable communication" [1,2].

The term "sustainable communication" is used to refer to any communication of issues related to corporate social responsibility while considering economic, social and environmental concerns on the one hand, and the interests of all stakeholders on the other [3,4]. In the face of a profusion of definitions and confusing names for this type of practice, [5] have highlighted three common meanings of sustainable communication, between "responsible communication", "eco-design of communication products" and "communication on sustainable development".

In recent years, the nature of corporate communication has changed. Social communication has become more focused and strategic in its execution. To that end, there remains a need for a deeper understanding of the underlying processes that drive returns from sustainable communication. Today, sustainable communication is a strategic lever on which companies rely to establish their legitimacy [6] and develop their brand equity [7]. It has become an essential means for organizations to engage in dialog and participatory communication with their stakeholders. Thus, this societal expression can be carried out via traditional channels (posters, events, . . . ) or by considering the website and social media as supposed to ensure a high level of attractiveness and interactivity [8,9].

Considering the phenomenon of greenwashing, recently widespread in society, and because of the proliferation of allegations of a societal nature, consumers have become

more suspicious and skeptical about the veracity of companies' social disclosures and the motivations of the latter in this regard [10]. To explain the different reactions of consumers to sustainable communication campaigns, the literature values the crucial role played by the attribution process in the treatment of societal information by the consumer [9,11].

By focusing on the consequences of assignments, several studies have pointed out that assigning an action to external circumstances or to the actor may influence the subject's attitude towards the actor and the trust placed in him [12]. The perceived motivation of consumers is now considered a key element in assessing and building brand equity through sustainable communication activities [13].

By relying on the theory of attribution, which is concerned with the explanations given by the individual for the acts he observes or of which he is a player, and by considering sustainable communication as a signal sent by the company, through this article we will try to better understand how consumers perceive and interpret this signal depending on the nature of sustainable communication.

Sustainable communication campaigns are most often social or environmental. However, in both scenarios, they may relate to activities that may or may not be related to the primary role of the enterprise, hence the need to address the issue of congruence. Many studies have examined the effects of congruence on several aspects, including word of mouth [14], brand loyalty [15] and consumer reactions to corporate social responsibility [16,17]. Considering that not all acts can be interpreted by consumers in the same way, we are interested in this research in the question of congruence between sustainable communication and business, to analyze its impact on the nature of perceived motivation by the consumer.

## 2. The Method

To respond to our problems to what extent can the nature of sustainable communication influence the perceived motivation of the company? Using the last two decades' major works on the sustainable communication issue linked with the attribution theory, we will try to draw up a summary of the research conducted in this direction. The selection of the articles and related works is based on the presence of the keyword "sustainable communication" in the title of a research paper in the Science Direct database. The results provided us with 15 results for the research papers with "sustainable communication" in the title and 46 results with the keyword "sustainable communication" in the title, abstract or author-specified keywords since the year 2000. We see the authors for these two search results in Figures 1 and 2. According to our model, we took in consideration all the works in the attribution theory frame.

The strategy and the model of the literature research are shown in Figure 3.

First, we will present and explain, in the light of attribution theory, the distinct types of motivation perceived by the consumer regarding sustainable business communications. Subsequently, the effect of the congruence between sustainable communication and the principal competence of the company on the nature of perceived motivation will be discussed to clarify the role of perceived effort and other moderating variables in this relationship. Management interests, limitations and research paths will eventually be presented.

It should be noted that we have relied on the Feldman and Lynch (1988) [18] model of accessibility and diagnostic value of information to consider that judgments of perceived motivations for engaging in sustainable activities can be transferred to perceived motivations for sustainable communication.

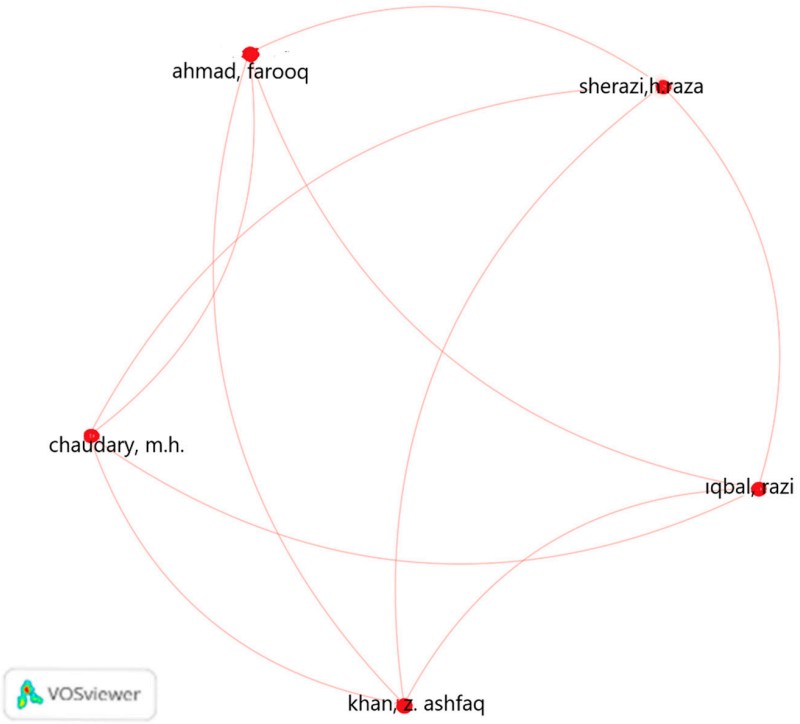

**Figure 1.** The authors of 15 works with keyword "sustainable communication" in the title. Source: VOSviewer.

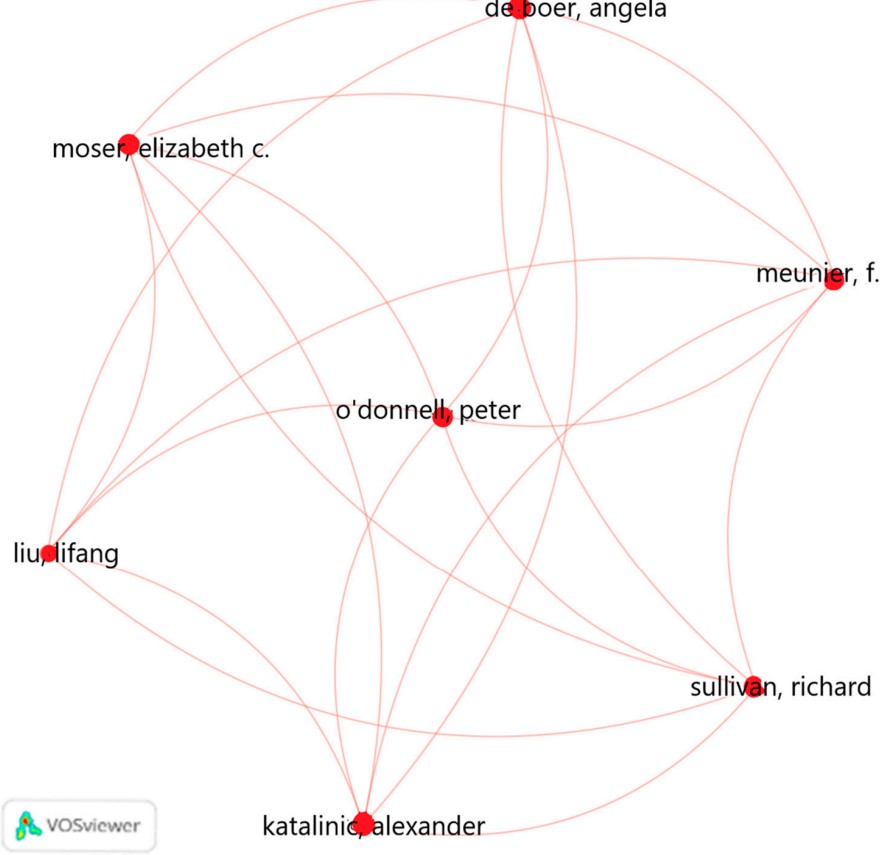

**Figure 2.** The authors of 46 works with keyword "sustainable communication" in the title, abstract or author-specified keywords. Source: VOSviewer.

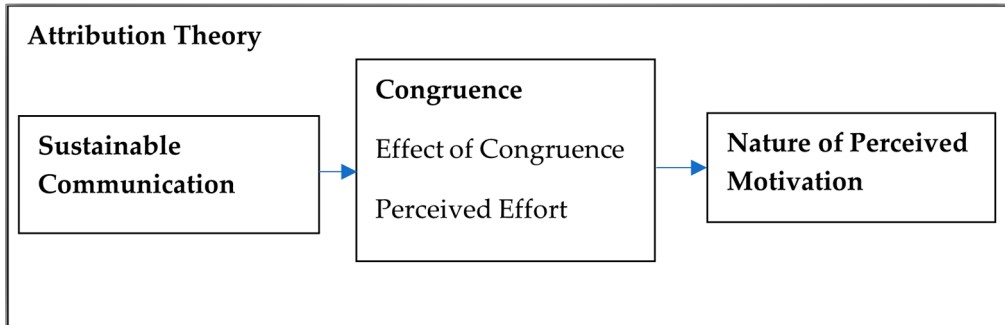

**Figure 3.** The Flowchart of the Model of the Literature Research. Source: Author's construction.

## 3. The Literature

### 3.1. The Perceived Motivation of the Sustainable Communication: Attribution Theory Approach

Attribution theory deals with how the social perceiver uses information to arrive at causal explanations for events. It examines what information is gathered and how it is combined to form a causal judgment.

One of the key factors influencing how consumers respond to communication campaigns and evaluate businesses is consumer skepticism [17,19]. In the context of societal initiatives and sustainable commitments, a misunderstanding of the corporate motivations behind these acts may be the source of this skepticism. The latter refers to consumer distrust of societal actions, communications extolling the company's good practices and the company's motivations in this regard [17,20].

In order to explain the divergence in consumer reactions to socially responsible business initiatives, the literature highlights the importance of consumer perceptions in relation to the motivations of the businesses behind these activities [16,21]. This then consists of knowing whether the consumer links the company's actions to selfish motives or public-interest motives [22].

Previous research has shown that consumers have a spontaneous curiosity about understanding the motivations of companies behind their long-term commitments [16], hence the interest in inferences [23]. Attribution theory provides an appropriate framework for dealing with this topic, as people express great interest in justifying why companies engage in sustainable practices and do not really trust them.

Originally developed by [24], the theory of attribution aims to explain how a person explains the events he or she observes or acts upon. It discusses how individuals attribute causes to events and how this cognitive perception consequently affects their subsequent attitudes and behaviors [25]. It is thus by attribution that this theory holds that causal analysis is inherent in people's need to understand, interpret events and organize their perceptual fields.

According to [24], the cases envisaged are divided into two main types; causes relating to the actor's own internal arrangements or to factors specific to the environment, known as external factors [26]. Previous research has been based on the principle of attribution theory to explain the positive or negative evaluation of sustainable business communication [27]. They pointed out that behavior attributed to intrinsic motives implies that consumers can judge these communication practices as a genuine business approach, stemming from its desire to be transparent and to improve its practices for the benefit of the community. On the other hand, behavior attributed to the actor's search for a reward (extrinsic motive) indicates that this act would not have been conducted in the absence of the reward. It, therefore, is motivated by opportunism.

Based on the same principle of causal inferences, new research has addressed the issue of perceived motivations behind corporate actions, this time distinguishing between the responsibilities related to the company's reasons for service, which focus on potential benefits for the business itself, and those related to public service reasons, which focus on the potential benefits to outsiders of the company [17,28,29]. Usually, the patterns in

business services are perceived negatively by consumers, as they signify an individualistic or opportunistic perspective. Public service motives are more favorably perceived as selfless and socially interested [30].

In the same context, recent research supports the complexity of societal initiatives. This is by mentioning that consumers are likely to identify multiple likely causal inferences for sustainable business engagement [31]. They suggest that in addition to traditional motivations of personal or public interest, four distinct types of causal inferences may occur: selfish motives, values-based motives, strategic motives and stakeholder motives [32,33].

Self-motivated assignments refer to perceptions of exploitation rather than corporate advocacy of the societal cause [32,33]. By attributing opportunistic and highly profitable motivations to the company's sustainable commitment, consumers perceive the sustainable communication campaign in question as an attempt to mislead them about the ethics of the company. They perceive a contradiction between these motivations and the societal cause principle, as the company is concerned with its own interests [17,33].

It should also be noted that, at this point, people perceive consumers as more trustworthy when they are willing to pay relatively high premiums for environmentally sustainable products. The reason is that those willing to pay high premiums are likely motivated by values rather than self-interest [1]. This finding, from an incentivized behavioral experiment, might transfer to people's perceptions of firms. Only when the engagement of a firm is strong or costly is it credible [34].

Values-based motivational assignments refer to linking the company's cause of sustainable engagement to its own moral, ethical and social norms [32]. In this case, consumers believe that the company cares about the cause and really cares about societal issues. These values-based assignments suggest that the firm's long-term commitments represent its genuine desire to operate within the favor of society [30].

Strategic reasons refer to the belief that the company can achieve its business objectives while supporting the cause [32,33]. However, consumers may be puzzled by the belief that the company is achieving its survival goals by undertaking and encouraging societal initiatives. On the one hand, these responsibilities may appear legitimate and tolerable to them, since any business must ensure the viability of its economy. On the other hand, profit-driven societal acts reflect behavior that results from economic rather than moral reasoning [33]. As a result, consumers may be outraged by the introduction of this profit-seeking interest in the context of societal initiatives that must be based primarily on values [35,36].

Stakeholder reason assignments reflect assumptions that the company is engaging in sustainable initiatives to meet stakeholder expectations [33]. The idea is that the company is adopting these practices in response to pressure from various interest groups, which may lead to negative connotations about the company's commitment and sustainable communication. These allocations are considered by consumers to be inconsistent with the true values and principles of the business [37] and merely a means of receiving rewards or avoiding accusations from stakeholders [32,33].

This literature review on perceived business motivation highlights the different types of causal attribution that can be inferred by consumers to explain the business motives behind its sustainable communication campaigns. Indeed, the nature of the motivation perceived by the consumer plays a decisive role in their subsequent relationship with the company or brand. It may have an effect on their brand assessment [7], their attitude towards the company [38], the perceived sincerity of communication and brand capital [11]. Moreover, the perception of the company's motivation may be influenced by various elements, whether they are related to the company and its commitment or related to the consumer.

Many studies have linked how consumers respond to socially responsible business commitments to the congruence between societal initiatives and business [30,39]. In the following, we will focus on the traditional distribution presented by attribution theory between internal and external motivation to address the effect of the perceived congruence

between sustainable communication and business activity on the nature of perceived motivation and to highlight the factors involved in this relationship.

### 3.2. The Effect of the Congruence between Sustainable Communication and Business on the Perceived Motivation

The literature shows a convergence of work on the existence of an impact of the perceived congruence between sustainable communication and the main competence of the company on the perceived motivation of the company and ultimately on the evaluation of the latter. However, there is no consensus on the nature of this impact. The results regarding consumers' interpretation of this congruence remain mixed.

Research suggests that in some situations where businesses develop communication campaigns about societal activities, consumers perceive selfish motivation of the business and the initiative in question are linked, while true altruistic behavior is attributed when business activity and the underlying cause do not match [15,40,41]. Initiatives outside the main scope of the business are more favorably valued by consumers as they interpret the underlying motivations to remain far from the pursuit of benefits and are linked to the well-being of society [41,42].

On the other hand, research has supported the positive relationship between congruence and consumer responses. It indicates that results are better for communication campaigns that are aligned with the company's core business and societal initiatives [27,43–45]. Indeed, the theory of associative networks [46], which focuses on the effects of sponsorship activities, emphasizes that consumers consider events to be more appropriate because they are highly consistent with the business. This can result in more favorable causal allocations that can improve consumer attitudes towards businesses.

Similarly, the congruence theory [47] states that people remember and prefer harmony and continuity in their thoughts and try to avoid conflicting thoughts. That is, the individual appreciates, by nature, the coherence between the new information and those already classified in their thoughts. More recent work has shown that the principle of congruence is valid for different research fields, and even for advertising [48,49] and client referral programs [50]. As for the context of the company's societal initiatives, the results suggest that consumers evaluate associations while trying to assimilate differences with their thoughts and attribute more credibility to initiatives that demonstrate some congruence [42].

In the case of strong congruence between the sustainable communication campaign and the business, it is reasonable to expect that the information will be easily integrated into the consumer's cognitive structure. The latter, which links this act to reasons of continuity in the business perspective of the company, will not deduce selfish motives from it. For example, the lack of congruence has been shown to require a reinterpretation of communication in order to create a link with the consumer. This can hinder the process of awarding the consumer and may contribute to the formation of questionable beliefs about the motives and interests of the business [43].

Supporting work shows a positive effect of the congruence between sustainable communication and business on the attribution process and the motivational nature perceived by the consumer. The effect of perceived effort and moderating variables in this direction will be discussed below.

The Role of Perceived Effort

In the case of sustainable communication about an activity or a cause related to the main competence of the company, several operational levers (information, material and organizational resources) appear to be exploited. However, when it is non-business communication, it is assumed that the business can only provide financial resources, which translates into less effort for consumers than investing energy and time [16].

In the same sense, [11] suggests that the nature of sustainable communication informs the degree of commitment of the firm, and that a weakly congruent cause implies a limited commitment to the financial aspect. They emphasized the need for consumers to perceive

less effort from the company in the case of a sustainable communication campaign that is not congruent with the case of a congruent cause.

Considering the principle of attribution theory [51], the actor's efforts are attributed to an internal nature while luck is an external cause. Perceived effort is then supposed to favor an internal attribution of motivation because it requires the intervention of will on the part of the subject. More specifically, the perceived effort is seen by consumers as a signal sent by the company, in the context of its sustainable communication, expressing its level of commitment to the societal activity in question [11]. The latter maintains that a strong perceived effort reflects a real willingness of the company to operate for the benefit of the sustained cause, which refers to the perception of an internal motivation.

In manipulating the perceived degree of congruence between sustainable communication and the main competence of the company (high congruence vs. low congruence), exploratory studies, carried out to explicitly test the impact of congruence on the interpretations and reactions of consumers with regard to communication campaigns and the company itself, have shown the moderating role of the social consciousness variable of the consumer at the level of the attribution process [11].

They imply that consumers with a strong societal awareness seem to be more involved in questioning the available information about the nature of the commitment provided by the company. Consequently, the influence of the congruence between the sustained cause and the company on the perceived effort of the latter is greater among those who have a strong societal awareness, which contributes to the perception of an internal motivation of the company behind its speaking. However, other people, without a societal consciousness, are less involved in interpreting the degree of perceived congruence between the company and its societal expression.

Societal consciousness then appears as a moderator in the relationship linking the congruence between sustainable communication and the company with the perceived internal motivation, its influence is carried out through the perceived effort. Moreover, the results indicated that neither congruence nor the consumer's societal awareness have a significant influence on perceived external motivation.

Indeed, the two notions of external and internal motivation can vary in the same direction. They are distinct and do not constitute contradictory elements in the mind of the consumer.

One cannot discuss consumer interpretations of sustainable business communications without addressing the effect of skepticism. Previous studies have shown that skepticism plays a role in explaining consumer behavior towards societal initiatives [22,34]. Indeed, a consumer who is more skeptical about issues of corporate social responsibility can question and examine, in more detail, any initiative taken in this direction. With respect to this more suspicious type of consumer, some research has shown that when expectations are not met, the consumer spends more time analyzing the actor's reasons behind these practices, resulting in more negative assignments [52].

A recent experimental study [53] highlighted the moderating effect of the skepticism variable in explaining consumer reactions to sustainable communication in a consistent scenario between the initiative in question and the business activity. The results show that the positive influence of congruence is only apparent among consumers with little skepticism about societal activities.

## 4. Results

The interpretation of sustainable corporate communication campaigns differs from person to person, depending on the perceived motivation of the company behind these acts. Varied reasons can be attributed by the consumer to these practices. They may share the traditional division between internal and external motivations, personal and public interest, or the more specific one, distinguishing between selfish, strategic, corporate or stakeholder-related reasons [54].

This difference in causal inferences may be due to situational and experiential factors related to consumers or other factors related to the firm, such as its societal reputation or the nature of the underlying cause that is the subject of this research.

Indeed, the literature previously presented shows that the perceived congruence between sustainable communication and business is interpreted differently by consumers. This is attributed by some to altruistic behavior, reflecting internal motivation, while by others it is linked to opportunistic and selfish behavior, reflecting external motivation. However, the positive impact of the congruence between the underlying cause and the firm on the perceived motivation of the firm is only apparent among consumers who are less skeptical of societal information and have considerable societal awareness. They are more involved in interpreting the company's sustainable communications and evaluating the efforts made to do so.

## 5. Conclusions

Companies are moving towards greater interaction with consumers to understand their needs, to offer them the best products and services possible and to achieve effective communication with them. Accordingly, it is particularly important to understand how consumers interpret different communication campaigns [55].

Based on previous research and attribution theory, this research contributes to understanding how consumers interpret and perceive sustainable corporate communications campaigns. In this study, a literature review was conducted to respond to the call by researchers and practitioners for a deeper theoretical and practical understanding of how to create effective sustainability communications based on attribution theory. A systematic review is a particularly appropriate way to study the effectiveness of interventions and to account for the conditions that frame sustainability communication.

To contribute to a more grounded scientific understanding of sustainability communication, this study proposes an interdisciplinary research agenda that considers the attribution theory, the effect of congruence and related variables and the nature of the perceived motivation. These umbrella concepts can be more fruitfully explored now that we know what exactly may be found under them.

Emphasizing the positive effect of long-term communication with business on the nature of perceived motivation, on the perception of internal motivation through the perceived effort of the company. Our research highlights, for managers, the moderating role played by each of the variables of societal sensitivity and skepticism in the consumer's perception of sustainable communication and its interpretation of the company's motivations. Such research will help identify the causal determinants of behavior and produce more effective sustainability messages. Practitioners could draw upon insights into congruence factors and understand the relative importance of sustainability attributes.

While perceived congruence plays a remarkable role in a favorable interpretation of sustainable communication, it is not a condition for its effectiveness. Through this research, we have highlighted the major effect of the perception of an effort on the part of the company on the nature of the motivations attributed to the latter.

At the managerial level, it would be recommended for companies wishing to develop their sustainable communication campaigns to identify the signals likely to influence consumer perceptions, to favor those having a favorable effect. They should also give more weight to the elements that may raise consumer skepticism. In the case of non-control, the latter can hinder all the efforts of the company. In addition, it would be important to look at the various elements that could promote the perceived effort of the company in the context of societal initiatives. Accordingly, as suggested in [1], strong and costly engagements by firms should be communicated to better motivate the consumer, decrease the skepticism and to increase credibility.

This review is subject to some limitations. The present study cannot fully avoid biases since we have limited our review to one database for a period of two decades.

However, it is a limitation of systematic reviews that they can only deliver partial solutions to practical problems.

In this research, the analysis of the link between the nature of sustainable communication and the perceived motivation of the company has been reduced to the question of congruence with the main activity of the company (congruent sustainable communication vs. non-congruent sustainable communication), which constitutes a limit for this research. When discussing the nature of sustainable communication, other elements can be used, such as the sustainable (social vs. environmental) axis of engagement, the source of communication (communication from the company itself vs. from an independent body) and the perceived motivation of each type of sustainable communication.

Finally, it must be added that there is a small consideration of sustainable communication. Assuming that companies are constituted through communication and largely based on communication processes, it is a necessity to set up sustainable communication to act economically, to guarantee the validity of the company's activities and, furthermore, to integrate companies into the changing society, thereby securing their future presence. It must be clearly emphasized that sustainability communication is an opportunity and not a tool to reduce costs.

**Author Contributions:** All authors equally contributed to the preparation of paper. Conceptualization, B.D. and C.D.; Formal analysis, B.D. and C.D.; Investigation, B.D. and C.D.; Methodology, B.D. and C.D.; Writing—original draft, B.D. and C.D.; Writing—review & editing, B.D. and C.D. All authors have read and agreed to the published version of the manuscript.

**Funding:** This work has been supported by the Scientific Research Projects Commission of Galatasaray University under grant number #FBA-2022-1082.

**Institutional Review Board Statement:** Not applicable.

**Informed Consent Statement:** Not applicable.

**Conflicts of Interest:** The authors declare no conflict of interest.

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
