# Peer review of "Sustainable Communication; Perceived Motivation and Nature of the Commitment"

_sustainability, doi:10.3390/su14159783_

Round 1

Reviewer 1 Report

Dear authors,

Thanks to you, I can confirm your manuscript. Before publishing your manuscript, Please reconfirm it carefully. 

Thanks and best regards,

Author Response

Dear Reviewer,

First of all, we would like to kindly thank you  for your time and valuable comments, according to your remarks we spellchecked all manuscript and read it again for possible improvements and now we confirm the manuscript in this state.

Thanks again for your time, Best Regards,

Reviewer 2 Report

Review sustainability-1850731

This paper fits well into the scope of the journal, and the authors have thoroughly improved the first version of the manuscript.

There are only two details left to do before publication.

Add some more specific information to the abstract. For example, what is the case number of the papers included in the literature review?

It’s nearly impossible to read Figures 1 and 2. Please use a larger font size.

Author Response

Dear Reviewer,

First of all, we would like to kindly thank you  for your time and valuable comments. According to your remarks;

  • we added few sentences to the abstract (total of 77 words)  to give some numbers.
  • we resized the figures 1 and 2 to make it possible to read easily.
  • we read our manuscript again for possible improvements and now we confirm the manuscript in this from.

Thanks again for your time, Best Regards,

This manuscript is a resubmission of an earlier submission. The following is a list of the peer review reports and author responses from that submission.

Round 1

Reviewer 1 Report

Submitted paper represents an interesting and up-to-date consumer´s insight into the perception of sustainable communication of the company. This research study is written by the theoretical style of integrative review. Summarize existing knowledge in sustainability communication, motivation and commitment through a number of research studies and offers a detailed insight and a deeper understanding of this scientific or the application issue. The authors analyzed number of existing research studies and assessed the problem from various point of views. They also discussed the terms used, existing theories, methodological approaches and at the end they formulated their conclusions in the form of desirable generalization. 

Author Response

Dear Sir/Madam,

Thank you for your valuable and helpful comments, we''ve done the necessary minor spell check and improve the paper.

Thanks and a Happy New Year, Best Regards

Reviewer 2 Report

see file attached

Author Response

Dear Sir/Madam,

First of all thank you for your valuable and helpful comments and Happy New Year.

1- We've done the changes in abstract and Introduction part.

2- Added the necessary References that help our paper.

3- Added the point of view to the review and to the conclusions.

Best Regards,

Reviewer 3 Report

Thank you for reading the contribution manuscript. There seems to be no big problem with English sentences or logic. However, it seems that some supplementations are needed. 1. The reader may have difficulty reading because there is no research model or research hypothesis that this study aims for. It would be better to organize what we found in this study into a visual picture or table. 2. I think it will be a much more convincing study if you present continuous communication-related causal models and research hypotheses and try empirical analysis on them. 3. The conclusion also presents general content, so I think again about what implications can be obtained from practice. 
Thank you.

Author Response

Dear Sir/Madam,

Thank you for your valuable and helpful comments and have a Happy New Year.

1- added the figure and some explanations in the introduction to make it clear for the reader

2- we are limited to literature review linked with attribution theory in this work considering the congruence and perceived motivation, hope to do an empirical research in a following work. this literature review is a start for us to work on a series of research in this field.

3- we've improved the conclusions.

Best Regards, 

Round 2

Reviewer 3 Report

Dear Authors,

I read the revised manuscript well.
I confirmed that it has improved a lot compared to the last time. I look forward to good results and hope it will be of great help to the working-level staff.

Thank you very much.

Best regards,